# Integration of academic and health education for the prevention of physical aggression and violence in young people: systematic review, narrative synthesis and intervention components analysis

G.J. Melendez-Torres,[1] Tara Tancred,[2] Adam Fletcher,[3] Rona Campbell,[4] James Thomas,[5] Christopher Bonell[2]

For numbered affiliations see end of article.

**Correspondence to**
Dr G.J. Melendez-Torres;
melendez-torresg@cardiff.ac.uk

## ABSTRACT

**Objectives** To systematically review evidence on the effectiveness of interventions including integration of academic and health education for reducing physical aggression and violence, and describe the content of these interventions.

**Data sources** Between November and December 2015, we searched 19 databases and 32 websites and consulted key experts in the field. We updated our search in February 2018.

**Eligibility criteria** We included randomised trials of school-based interventions integrating academic and health education in students aged 4–18 and not targeted at health-related subpopulations (eg, learning or developmental difficulties). We included evaluations reporting a measure of interpersonal violence or aggression.

**Data extraction and analysis** Data were extracted independently in duplicate, interventions were analysed to understand similarities and differences and outcomes were narratively synthesised by key stage (KS).

**Results** We included 13 evaluations of 10 interventions reported in 20 papers. Interventions included either full or partial integration, incorporated a variety of domains beyond the classroom, and used literature, local development or linking of study skills and health promoting skills. Evidence was concentrated in KS2, with few evaluations in KS3 or KS4, and evaluations had few consistent effects; evaluations in KS3 and KS4 did not suggest effectiveness.

**Discussion** Integration of academic and health education may be a promising approach, but more evidence is needed. Future research should consider the 'lifecourse' aspects of these interventions; that is, do they have a longitudinal effect? Evaluations did not shed light on the value of different approaches to integration.

## Strengths and limitations of this study

► We used an exhaustive search including 19 databases and 32 websites.
► We used an innovative method to describe key components in this class of interventions.
► However, it was challenging to identify studies for inclusion.
► Meta-analysis was not possible because of the diversity of outcomes and raters.

and harm to young people and the wider society.[1 2] One UK study found that 10% of young people aged 11–12 reported carrying a weapon and 8% admitted attacking someone with intent to hurt them seriously.[3] By age 15–16, 24% of students reported they have carried a weapon and 19% reported attacking someone with the intention to hurt them seriously.[3] Early aggression and antisocial behaviour are strongly linked to adult violent behaviour.[4 5]

School-based health education can be effective in reducing violence.[6–8] However, school-based health education is increasingly marginal in many high-income countries, partly because of schools increasing focus on attainment-based performance metrics. In England specifically, health education is not a statutory subject,[9–11] and school inspectors have a limited focus on how schools promote student health.[12]

One way to avoid such marginalisation is to integrate health education into academic lessons. For example, health-related content can be seamlessly integrated into existing academic lessons or discrete additional health education lessons can also include academic

## INTRODUCTION

Violence among young people is a public health priority due to its prevalence

learning elements. This strategy may bring other benefits because: larger 'doses' may be delivered; students may be less resistant to health messages weaved into other subjects; and lessons in different subjects may reinforce each other.[13] [14] Conversely, those teaching academic subjects may be uninterested or unqualified to teach health topics. Though theories of change in this class of interventions are diffuse, one important way in which they could be effective is by promoting developmental cascades involving the interplay of cognitive and non-cognitive skills.[15] [16] Interventions integrating academic and health education could address violence by developing: social and emotional skills such as self-awareness, self-regulation, motivation, empathy and communication[17]; healthier social support or norms among students[15] [18] [19]; knowledge of the costs[20] and consequences[21] of substance use; media literacy skills to critique harmful media messages; and modifying students' social norms about antisocial behaviours.[13] [20] [22–24] Our work synthesising the theories of change underlying these interventions (Tancred *et al*, in press) identified that interventions aimed to integrate and thus erode boundaries between academic and health education, between students and teachers (so that relationships were improved and teachers might function more effectively as behavioural role models) and between classrooms and schools and schools and families (so that violence prevention messages communicated in classrooms might be reinforced by messaging in other settings).

Despite policy interest in these interventions, they have not previously been the subject of a specific systematic review. Previous systematic reviews have focused on socioemotional learning interventions or school-based interventions generally,[6–8] without considering interventions that specifically integrate with academic lessons as defined above. Our focus on violence is informed by preliminary consultation, scoping work and logic model development suggesting that violence is an outcome especially amenable to these interventions. In the present review, we examined the characteristics of interventions that integrate academic and health education to prevent violence, and synthesised evidence for their effectiveness. That is, our research questions were: what are the overarching features relevant to integration of interventions that integrate academic and health education, and are these interventions effective at different key stages (KS) in reducing physical aggression and violence?

## METHODS
This review was part of a larger evidence synthesis project on theories of change, process evaluations and outcome evaluations of integration of academic and health education for substance use and violence. We registered the protocol for this review on PROSPERO (CRD42015026464, https://www.crd.york.ac.uk/prospero/), and it is enclosed as online supplementary file 1.

## Inclusion and exclusion
Studies were included regardless of publication date or language. We included randomised controlled trials of interventions integrating academic and health education, the former defined as specific academic subjects or general study skills. We defined education as 'health education' seeking to improve the health and well-being of students (including social and emotional learning and other forms of violence prevention). We included school-based interventions that seamlessly incorporated health education into existing academic lessons and interventions that provided discrete health education lessons with additional academic components. Interventions could be delivered by teachers or other school staff such as teaching assistants, but may also have been delivered by external providers, for example, from the health, voluntary or youth service sectors. We did not include interventions solely addressing social conduct in the classroom; relationships with peers or staff; attitudes to education, school or teachers; or aspirations and life goals. Our definition also excluded interventions which: were delivered in mainstream subject lessons but did not aim to integrate health and academic education; trained teachers in classroom management without student curriculum components; or were delivered exclusively outside of classrooms, as these did not seek to integrate academic and health education. Interventions focusing on targeted health-related subpopulations (eg, children with cognitive disabilities) were excluded as we were interested in universal interventions.

For this review, we focus on violence outcomes, defined as the perpetration or victimisation of physical violence including convictions for violent crime. While we preferred direct measures of physically violent and physically aggressive behaviours, we included outcomes that were a composite of physical and non-physical (eg, verbal or emotional) interpersonal violence, but excluded composite measures that also included items not focused on interpersonal violence, such as damage to property.

## Search strategy
In our original search, undertaken between November and December 2015, we searched 19 databases and 32 websites, and contacted subject experts (see online supplementary file 2 for full details). We subsequently updated our search in February 2018 using PsycINFO and CENTRAL, as all of our original study hits were recovered from these databases.

## Study selection
Pairs of researchers double-screened titles and abstracts in sets of 50 references until 90% agreement was reached, with disagreements discussed at every stage. Subsequently, single reviewers screened each reference. We located the full texts of remaining references and undertook similar pairwise calibration with disagreements discussed, followed by single screening. Reports were translated into English where necessary. Using an existing tool,[25] we

extracted data independently in duplicate from included studies and assessed trials for risk of bias using a modified version of the Cochrane assessment tool.[26] Authors were contacted where study data were missing.

## Synthesis methods

We undertook an intervention components analysis.[27] This was undertaken inductively by one researcher and audited by two other researchers, and used intervention descriptions to draw out similarities and differences in intervention design using an iterative method. Intervention descriptions were read and reread and then coded manually. The goal of this analysis was to use a set of descriptors to characterise aspects of the integration of academic and health education in the intervention. Intervention descriptions were rarely detailed enough to permit 'deep' engagement with the specific content of the interventions provided in included evaluations. The intervention components analysis identified overarching domains that accounted for similarities and differences between interventions in their integration of academic and health education, and developed within each domain a set of overlapping categories that described these similarities and differences. Finally, we synthesised outcomes narratively due to the heterogeneity in included outcome measurement. We categorised the timing of intervention effect by period of schooling, defined in terms of English schools' KS system. KS1 includes school years 1–2 (age 5–7 years), KS2 includes years 3–6 (age 7–11 years), KS3 includes years 7–9 (age 11–14 years), KS4 includes years 10–11 (age 14–16 years) and KS5 includes years 12–13 (age 16–18 years).

We could not formally assess publication bias because heterogeneity in outcome measurement precluded meta-analysis.

## Patient and public involvement

Because this review focused on public health interventions that were generally preventive in nature, patients were not involved per se. However, stakeholders were extensively consulted in the development of research questions and in assessing the implications of the findings. In addition, findings were disseminated via stakeholder events, and a series of one-to-one consultations took place to ensure the relevance and salience of study findings.

## RESULTS

In our original search, we found and screened 76 979 references, of which we retained 702 for full-text screening and were able to assess 690. Of 62 relevant reports included in the overall project, 10 evaluations of eight interventions were reported in 14 papers that considered violence and are reported in this review. Our update search yielded 2355 references, of which we retained 41 for full-text screening and included six papers

reporting three evaluations (figure 1). This yielded a total of 13 evaluations reported in 20 papers.

## Included studies and their quality

All trials randomised schools except the Bullying Literature Project, which randomised classrooms (table 1). All evaluations were conducted in the USA, except for Gatehouse,[28] which was an Australian study, and Learning to Read in a Healing Classroom,[29 30] which took place in the Democratic Republic of the Congo. All control arms consisted of education-as-usual or waitlist controls, though Second Step[31–33] offered a brief antibullying intervention with low take-up.

Interventions were diverse and are summarised below in the intervention components analysis. Only two interventions (Bullying Literature Project,[34] Youth Matters[35]) were wholly delivered by external staff. Several (Gatehouse,[28] Positive Action,[36] Steps to Respect[37]) linked classroom-based delivery to school-level work to support and reinforce implementation. Promoting Alternative Thinking Strategies (PATHS)[38] and Reading, Writing, Respect and Reconciliation (4Rs)[19] also emphasised teachers' professional development.

Evaluation quality varied (table 2). Appraisal was hampered by poor reporting of some aspects of trial methods. Only four studies reported evidence of low risk of bias for random generation of allocation sequence; the remainder were unclear. Only one study reported information on concealed allocation. In Linking the Interests of Families and Teachers (LIFT),[39] outcome assessors were blinded, resulting in low risk of bias in this domain, but all other interventions were of unclear risk of bias. All interventions included reasonably complete outcome data, and in only one evaluation did unit of analysis issues pose a risk of bias. In some studies such as Steps to Respect, follow-up was shorter than intervention length. Evaluations also differed in size, ranging from 7 classrooms to 63 schools.

## Intervention components analysis

This identified four themes describing included interventions: approach to integration, position of integration, degree of integration and point of integration. Included interventions are described in table 1, and the components analysis is summarised in table 3.

### Approach to integration

Interventions approached the rationale for and strategy of integration in different and overlapping ways. These overlapped across interventions, but were not mutually exclusive, and described the types of academic foci that interventions used to integrate academic and health education. Several (4Rs, Bullying Literature Project, Steps to Respect, Youth Matters) focused on *literature* as a focus for integration, using children's books as a prompt for social–emotional learning. These interventions targeted language arts or literacy lessons as an opportunity to provoke discussion, role play and model positive strategies

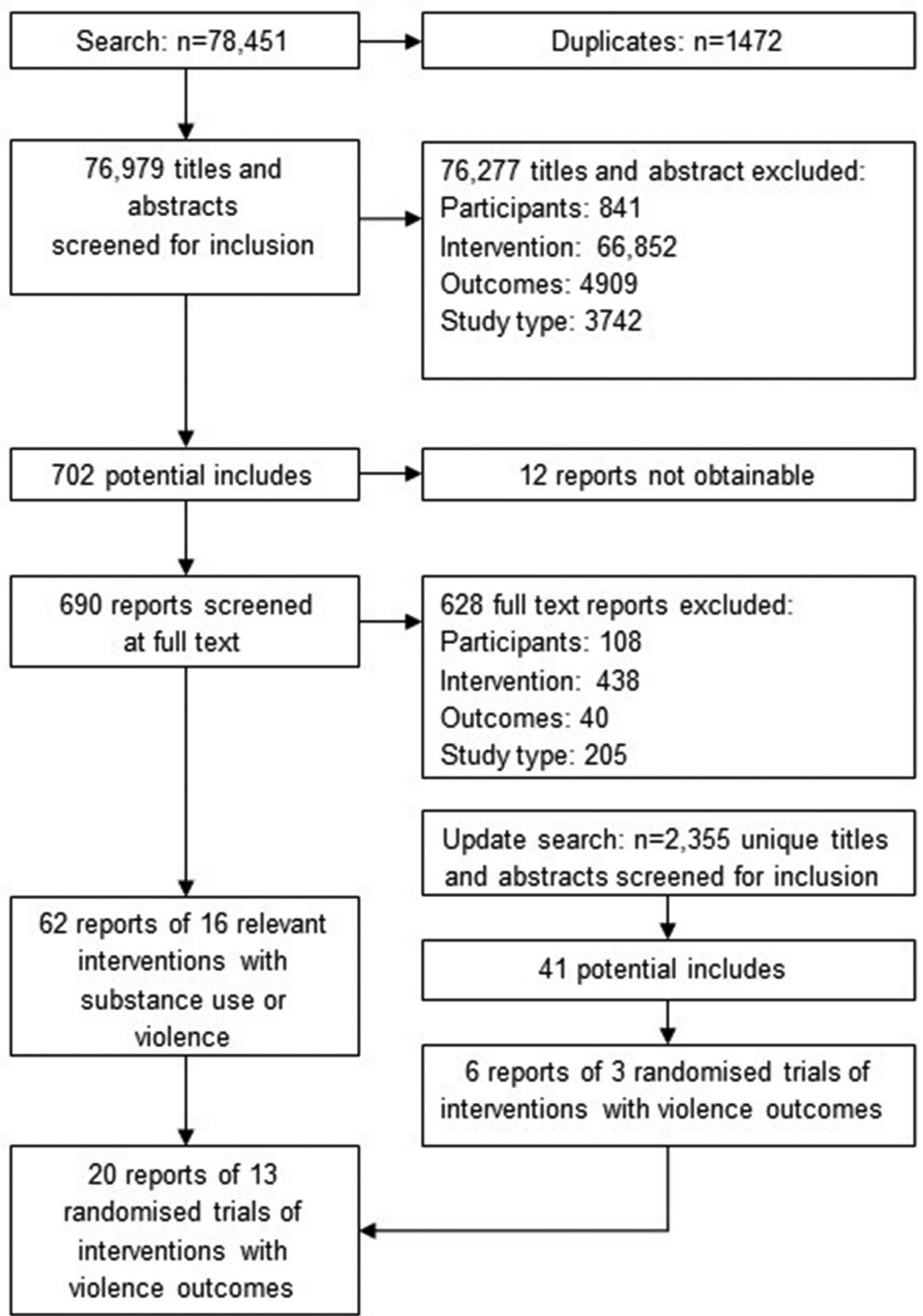

**Figure 1** Preferred Reporting Items for Systematic Reviews and Meta-Analyses flowchart.

to avoid violence. Gatehouse explicitly used a 'critical literacy' approach to inspire reflection on programme lessons in English classes. Another approach to integration emphasised *local development*, where interventions supported teachers to link health education across academic subjects in each school in a 'local' fashion. For example, in PATHS, teachers received suggestions on how to integrate programme learning across English, history and social studies lessons, while in Second Step, this was an encouraged aspect of classroom delivery. In both cases, teachers received guidance and support to integrate health education messages into academic education, but were given substantial latitude to determine how and when to do this in the school day. A third approach was

**Table 1** Characteristics of included studies

| Evaluation, setting and studies | Sample characteristics | Intervention description | Control description |
|---|---|---|---|
| Bullying Literature Project California, USA Couch[34] | Four classrooms, 95 teacher reports, 90 students (IG); three classrooms, 55 teacher reports, 42 students (CG) Students enrolled in years 4 and 5, followed up 1 week postintervention 42.8% girls, 57.2% boys 9.6% African American, 63.3% Hispanic, 9.0% Caucasian, 3.0% Asian, 4.2% other, 10.2% did not report >50% of students received free or reduced-cost lunch | This intervention aims to reduce bullying by introducing themes related to bullying through children's literature. It also provides an opportunity for children to role model practical skills to address or avoid bullying. The Bullying Literature Project integrates themes related to bullying into the children's literature used within a standard English curriculum. Students then had the opportunity to practice and reinforce skills via writing activities. The intervention was delivered by school psychologists supervised by the PI and lasted over 5 weeks in one school term. Additionally, the version including moral disengagement discussed the role of moral disengagement in each lesson as well. | Waitlist control |
| Bullying Literature Project—Moral Disengagement California, USA Wang[41] | Two classrooms, 42 students (IG); 2 classrooms, 42 students (CG) Students enrolled in year 4, followed up 1 week postintervention 53.6% girls, 46.4% boys 2.4% Asian, 3.6% Caucasian, 94% Hispanic SES details not reported | | Waitlist control |
| Gatehouse Melbourne, Australia Bond et al[28] | Two districts, 12 schools, 1335 students (IG); two districts, 14 schools, 1343 students (CG) Students enrolled in year 9, followed up for 3 years 53.2% girls, 46.8% boys 87.5% Australian-born 79.2% from two-parent family; 24.2% speak language other than English at home | Through teaching a curriculum (including integration of cognitive behavioural principles in English classes) and establishing a school-wide adolescent health team, Gatehouse aims to: build a sense of security and trust in students; enhance skills and opportunities for good communication; and build a sense of positive regard through participation in school life. The intervention was delivered by teachers over the course of 2 school years, supported by the school-wide adolescent health team and by external consultants who themselves were experienced teachers. Integration was achieved by using English classes to convey cognitive behavioural techniques for self-management, including via a 'critical literacy' approach that uses poetry, literature, song, film and visual materials. | Education as usual |
| Learning to Read in a Healing Classroom Democratic Republic of the Congo Torrente et al[30] Aber et al[29] | 20 districts, 33 schools (IG); 19 districts, 30 schools (CG); 3857 students overall Students enrolled in years 3–5, followed up for 1 year 48% girls, 52% boys | The intervention is delivered over the course of a year and is designed for use in postconflict reconstruction areas. Teachers are supported to integrate social–emotional learning in literacy lessons, supported by a bank of lesson plans relating to reading and writing. Teachers additionally received substantial professional development, including 'teacher learning circles', and developed strategies to improve the learning environment. | Waitlist control |
| Linking the Interests of Families and Teachers (LIFT) Pacific Northwest, USA Reid et al[39] | Three schools, 214 students (IG); three schools, 147 students (CG) Students enrolled in year 6 and followed up over 7 years 49% girls, 51% boys 86% White, 14% ethnic minority 12% mothers less than high school graduate, 8% father less than high school graduate; 36% mother unemployed, 10% father unemployed; 22% single-parent families; 18% receiving benefits; 20% less than $15 000/year in the early 1990s | Classroom instruction and discussion on specific social and problem-solving skills followed by skills practice, reinforced during free play using a group cooperation game with review of behaviour and presentation of daily rewards. There is also a parent evening to engage families and opportunities for parents to engage with teachers. The intervention was delivered by teachers and special instructors. Integration was achieved by teaching study skills alongside social–emotional education content, and was delivered over the course of 1 school year. | Education as usual |

Continued

**Table 1** Continued

| Evaluation, setting and studies | Sample characteristics | Intervention description | Control description |
|---|---|---|---|
| Positive Action Chicago Chicago, USA Li et al[36] Lewis et al[45] | Seven schools,~240 students (IG); seven schools,~260 students (CG) Students enrolled in year 4 and followed up over 6 years ~48% girls,~52% boys 55% African American, 32% Hispanic, 9% White non-Hispanic, 4% Asian, 5% other or mixed 83% receiving free lunch | Teachers provide lessons covering six units: self-concept; positive actions for mind and body; positive social–emotional actions; managing oneself; being honest with oneself; and continually improving oneself. Content includes 140 lessons per grade per year from years 1 to 13. In addition, an implementation coordinator and school climate team are appointed to support the intervention. The intervention is primarily delivered by teachers and school staff; in both trials, this was supported by extensive professional development and training. Integration was achieved by linking academic learning to social–emotional and health-related learning, for example, by including content on problem solving and study skills alongside positive actions for mind and body, and by encouraging teachers to reflect Positive Action content in academic lessons. | Education as usual |
| Positive Action Hawaii Hawaii, USA Beets et al[43] | 10 schools, 976 students (IG); 10 schools, 738 students (CG) Students enrolled in years 2 or 3 and followed up over 4 or 5 years 50% girls, 50% boys 26.1% Hawaiian, 22.6% mixed, 8.6% White, 1.6% African American, 1.7% American Indian, 4.7% other Pacific Islander, 4.6% Japanese, 20.6% other Asian, other 7.8%, unknown 1.6%. Control schools had on average 55% free/reduced lunch students, whereas intervention schools had on average 56% free/reduced lunch students | | Education as usual |
| Promoting Alternative Thinking Strategies (PATHS) Minnesota and New York State, USA Crean[38] | Seven schools, 422 students (IG); seven schools, 357 students (CG) Students enrolled in year 4 and followed up over 3 years 57% girls, 43% boys 51% White, 38% African American, 10% other, 17% Hispanic 33% from single-parent homes; 39% families with income less than $20 000/year, 43% below the federal poverty line; 11% no parent with high school diploma | An intervention to reduce conflict by improving students' social–emotional and thinking skills through a curriculum (including study skills), the establishment of a positive classroom environment and generalised positive social norms throughout the school environment. Lessons are grouped into three units addressing readiness and self-control, feelings and relationships and interpersonal problem solving. These units cover five domains: (1) self-control; (2) emotional understanding; (3) positive self-esteem; (4) healthy relationships; and (5) interpersonal problem-solving skills. The intervention is delivered by teachers supported by consultants, with 131 lessons delivered over 3 years (two to three times per week, 20 to 30 min each). Integration was achieved by linking study skills to social–emotional learning, by supporting teachers to include children's literature in reinforcing concepts, and by providing ideas to link PATHS to English, social studies and history lessons. | Education as usual |
| Reading, Writing, Respect and Reconciliation (4Rs) New York City, USA Jones et al[19] Jones[15] | Nine schools, 515 students (IG); nine schools, 427 students (CG) Students enrolled in year 4 followed up for 3 years (only results up to 2 years available) 51.2% girls, 48.8% boys 41.1% African American, 45.6% Hispanic, 4.7% Caucasian, 8.6% other 31% low parental education, 15.1% parental unemployment, 53.4% single-parent household, 61.8% living in poverty | This intervention includes two components: (1) a seven-unit, 21–35 lesson literacy-based curriculum in conflict resolution and social–emotional learning for children in primary school (from year 1 to year 6); and (2) intensive professional development for teachers. The intervention was delivered by teachers after this extensive professional development. Integration was achieved by using literature as a springboard to help students understand anger and develop skills in listening, cooperation, assertiveness and negotiation. | Education as usual |

Continued

**Table 1** Continued

| Evaluation, setting and studies | Sample characteristics | Intervention description | Control description |
|---|---|---|---|
| Second Step Illinois and Kansas, USA Espelage et al[33] Espelage et al[32] Espelage et al[31] | 18 schools, 1940 students (IG); 18 schools, 1676 students (CG) Students enrolled in year 7, followed up yearly over 3 years 48.1% girls, 51.9% boys 26.4% African American, 24.7% Caucasian, 34.2% Hispanic, 14.7% biracial and all others 74.1% free or reduced lunch | This intervention includes 15 weeks of classroom lessons taught weekly or every 2 weeks throughout the school year for 3 years. Teachers are supported by professional development training to deliver intervention content, which includes bullying, problem-solving, emotional regulation and empathy, alongside videos. Teachers also receive plans to support integration of Second Step content into academic lessons. Modelling, role play and coaching are included in the intervention. Students receive homework to reinforce skills, and use group and collaborative work to practice skills. | Education as usual, with additional bullying resources |
| Steps to Respect I Pacific Northwest, USA Frey et al[37] | Three schools (IG), three schools (CG); 1126 students total Students enrolled in years 4 through 7; followed up for 1 year in endpoint-based analyses 49.4% girls, 50.6% boys 70.0% White, 9% African American, 12.7% Asian, 7.0% Hispanic, 1.3% Native American SES indices not stated | This is an antibullying intervention with both school-wide and classroom components. The school-wide components create new disciplinary policies for bullying and improve monitoring of and intervention in bullying. Classroom curricula teach positive social norms and improve social–emotional skills for better engagement with bullying. The intervention was delivered by classroom teachers alongside school-wide bullying policy teams. Biweekly lessons in the Steps to Respect curriculum are supported by 8–10 literature-based lessons presented over a 12–14-week period. This intervention integrates academic and health education by developing literacy skills alongside furthering understanding of the Steps to Respect curricular themes. | Waitlist control |
| Steps to Respect II North-Central California, USA Brown et al[42] | 17 schools (IG), 16 schools (CG); 2940 students total Students enrolled in years 4 through 6; followed up for 1 year 51% girls, 49% boys (IG); 48% girls, 52% boys (CG) 52% White, 7% African American, 6% Asian, 43% Hispanic, 35% other or mixed race (IG); 53% White, 6% African American, 6% Asian, 41% Hispanic, 35% other or mixed race (CG) School-level average of 40% on free or reduced-price lunch | | Waitlist control |
| Youth Matters Denver, USA Jenson[35] Jenson et al[46] Jenson et al[44] | 14 schools, 702 students (IG), 14 schools, 462 students (CG) Students enrolled in year 5 and followed up for 3 years 50.6% girls, 49.4% boys 59.1% Latinx, 14.7% African American, 16.8% American Indian, Asian American, or mixed, 9.3% Caucasian SES indices not reported | Youth Matters promotes the development of healthy relationships and social competency and the development of social resistance. Classroom discussions around social issues promote positive social norms. Over four modules with 10 lessons, delivered over 2 years, students read age-appropriate stories, receive social-emotional learning and practice skills. The intervention was delivered by educational specialists from outside the school. Integration was achieved by using 30–40-page stories in each module intended to support schools in meeting academic standards in academic and health education. | Education as usual |

IG, intervention group; CG, control group; SES, social-economic status; PI, principal investigator

**Table 2** Appraisal of included studies

| Intervention name | Random generation of allocation sequence | Concealed allocation | Blinding | Complete outcome data | Reporting not selective | Controlled for confounding | Accounted for clustering | Reduced other forms of bias | Suitable control group |
|---|---|---|---|---|---|---|---|---|---|
| Bullying Literature Project | NC | NS | NS | Yes | NC | NC | NS | NS | NC |
| Bullying Literature Project — Moral Disengagement | NS | NS | NS | Yes | Yes | Yes | No | Yes | Yes |
| Learning to Read in a Healing Classroom | Yes | Yes | NS | No | Yes | Yes | Yes | Yes | Yes |
| Linking the Interests of Families and Teachers (LIFT) | Yes | NS | Yes | Yes | NC | Yes | Yes | Yes | NC |
| Positive Action Hawaii | NC | NS | NS | Yes | NC | NC | Yes | Yes | Yes |
| Positive Action Chicago | Yes | NS | NS | Yes | NC | Yes | Yes | Yes | Yes |
| Promoting Alternative Thinking Strategies (PATHS) | NC | NS | NS | Yes | NC | NC | Yes | Yes | NC |
| Reading Writing, Respect and Resolution (4Rs) | Yes | NS | NS | Yes | No | Yes | Yes | Yes | Yes |
| Second Step | Yes | NS | NS | Yes | No | Yes | Yes | Yes | Yes |
| Steps to Respect I | NC | NS | NS | Yes | NC | NC | Yes | Yes | NC |
| Steps to Respect II | NC | NS | NS | Yes | NC | NC | Yes | Yes | Yes |
| Gatehouse | NC | NS | NS | Yes | NC | Yes | Yes | Yes | Yes |
| Youth Matters | NC | NS | NS | Yes | NC | Yes | Yes | NS | Yes |

NC, not clear; NS, not stated.

**Table 3** Key themes in the intervention components analysis

| Key theme | Components within theme | Bullying Literature Project | Gatehouse | Learning to Read in a Healing Classroom | LIFT | Positive Action | PATHS | 4Rs | Second Step | Steps to Respect | Youth Matters |
|---|---|---|---|---|---|---|---|---|---|---|---|
| Approach to integration | Literature: did interventions use literature and language arts as the key vehicle for delivery? Local development: did interventions support teachers to link health education across academic subjects in each school? Linking to developmental concerns: did interventions link academic education and personal health and development? | Literature: use of children's books as basis for lessons | Literature: English classes used as a key vehicle for delivery of cognitive behavioural content relating to emotional learning and self-regulation. Linking to developmental concerns: underlying the theory of change was a connection between school health and well-being and academic attainment, and the need to create 'healthy' schools | Literature: use of reading and language arts lessons to support socioemotional learning by providing lesson plans | Linking to developmental concerns: study skills were presented alongside socioemotional learning skills, such as empathy and how to play with peers. This content was restricted to the year 6 arm of the intervention | Local development: teachers are supported to integrate health education lessons (both social-emotional learning and health and well-being, eg, hygiene) throughout academic learning. Linking to developmental concerns: key aspect of theory of change is linking academic achievement with physical and mental health and well-being, character development | Linking to developmental concerns: content on social-emotional learning was presented alongside study skills in later years of the programme. Literature: English (but also history and social studies classes) was used as a key opportunity to reinforce concepts taught in discrete manualised social-emotional learning lessons | Literature: the intervention centres on a literacy-based curriculum relating conflict resolution and social-emotional learning to children's literature | Local development: social-emotional learning is integrated into academic lessons alongside a manualised programme of content relating specifically to bullying, problem-solving, emotional regulation content and multimedia resources | Literature: the classroom component of this intervention relates to a programme of literature-based lessons designed to convey antibullying messages | Literature: stories are used to discuss healthy relationships, resistance to bullying and aggressive behaviours, and to practice skills, including via projects relating to literacy lessons |
| Domains of integration | Classroom: did interventions focus on the classroom? Classroom and whole school: did interventions include whole-school change components alongside classroom components? Classroom, whole-school and external domains: did interventions also include parent engagement alongside classroom and whole-school components? | Classroom: focus on classroom teaching only via book reading and accompanying activities | Classroom and whole-school domains: in addition to classroom learning strategies, a school health team supported by external consultants sought to identify ways to improve school climate to promote health and well-being | Classroom and whole-school domains: in addition to lesson plans to support classroom teaching, pedagogic circles facilitated school meetings led to exchange of ideas on how to improve school climate | Classroom, whole-school and external domains: in addition to supporting study skills alongside social-emotional learning, parents received a series of parenting classes and teachers were encouraged to communicate with parents via a phone line recorded message | Classroom, whole-school and external domains: in addition to extensively manualised lessons, a school climate team was assembled as part of the intervention with a schoolwide 'champion' for intervention implementation. Parents are also involved through homework and 'take-home' assignments, as well as community engagement, though this was not a feature in the Chicago trial | Classroom, whole-school and external domains: manualised lessons relating to social-emotional learning and self-regulation are accompanied with school-wide implementation to promote generalised positive norms and parent information | Classroom: teachers receive substantial professional development to implement the intervention using specific materials prepared as part of the intervention | Classroom: intervention delivered in the classroom context specifically | Classroom and whole-school domains: in addition to classroom literacy-based learning, a whole-school policy team developed school-wide responses to bullying | Classroom and whole-school domains: lessons delivered in the classroom context, but whole-school events 'showcasing' work part of the intervention activities |
| Degree of integration | Did interventions include full or partial integration of health education alongside academic education? | Full integration: lessons designed to develop literacy skills | Full integration: the use of 'critical literacy' to convey social-emotional learning was seamlessly integrated into English classes | Full integration: lessons designed to integrate social-emotional learning into enhanced provision of reading and literacy | Partial integration: the intervention was set apart from other academic learning | Partial integration: discrete lessons relating to Positive Action are presented as part of the intervention | Partial integration: manualised intervention lessons presented alongside academic content | Full integration: learning is presented alongside literature and reading lessons | Partial integration: separate lessons for intervention content are presented alongside integration | Full integration: lessons designed to address key literacy goals | Full integration: intervention 'led' by literacy and literature content |
| Timing of integration | Were interventions 1 year or multiple years in duration? | 1 year | Multiple years | Multiple years | 1 year | Multiple years | Multiple years | Multiple years | Multiple years | Multiple years/1 year | Multiple years |

*linking to developmental concerns*, emphasising not so much the comprehensive integration of academic and health education but rather the inter-relationships between academic success and broader development, health and well-being. These interventions viewed academic education through a 'health' lens, in addition to viewing health education through an 'academic' lens. From a conceptual perspective, this meant that the inter-relationships between academic achievement, and student health and well-being were emphasised in theories of change. From a practical perspective, this meant that interventions paired activities such as study skills lessons with social–emotional learning (eg, in PATHS). For example, the theory of change underlying Gatehouse related to the creation of healthy social milieus in schools that would also support academic attainment; practically, this manifested as enhancement of academic lessons to improve interpersonal skills and emotional regulation. Similarly, Positive Action tied together individual student attainment with student health and well-being in their theory of change, with lessons focused on problem-solving and goal setting, among other topics.

### Domains of integration

Some interventions (4Rs, Bullying Literature Project and Youth Matters) were exclusively *classroom-focused* while others (Gatehouse, Steps to Respect) used *classroom and whole-school* strategies to reinforce and extend learning. For example, Gatehouse involved school implementation support teams, while Steps to Respect deployed a school-wide 'policy team' to revise and develop antibullying policies. Other interventions, (PATHS, Positive Action) used *classroom, whole-school environment and external domain* (parent information) strategies consistent with the health-promoting schools approach promulgated particularly by the WHO, which in the USA is known as the Comprehensive School Health Programme model.[40]

### Degree of integration

In some interventions, health education was fully integrated (woven seamlessly) into everyday academic lessons (Gatehouse, 4Rs, Youth Matters), while in partially integrated interventions, health education involved distinct lessons, although also covering academic learning (Positive Action).

### Timing of integration

Most interventions were multiyear, though two involved only 1 school year (LIFT, Bullying Literature Project).

### Intervention effects

Perpetration measures included bullying (physical or physical/verbal), aggression against peers and others and violent behaviours including injuring others. Measures involved different raters, including students, teachers and observers. Victimisation measures ranged from physical violence specifically to interpersonal aggression more generally. Heterogeneity of definition, measurement and form of effect sizes precluded meta-analysis. No included studies described effects for KS1 or KS5. Measures and corresponding effect estimates are included in table 4.

### Violence perpetration: KS2

Across the 10 evaluations reporting outcomes in this KS, effects were inconsistent, including within studies by rater.

In LIFT,[39] effects at the end of the first intervention year on observed physical aggression in the playground were similar for students with different levels of baseline aggression ($d=-0.14$ at mean, 1 SD and 2 SD above the preintervention mean); these findings being described as 'statistically significant'. However, after the first intervention year of 4Rs,[19] there were no effects on teacher-reported aggression (regression-estimated $b=0.02$, SE=0.05, based on a 1–4 scale). After the second intervention year,[15] there were effects on teacher-reported student aggression ($d=-0.21$, p<0.05). The Bullying Literature Project also reported no effects on physical aggression rated by teachers for individual students (IG [intervention group]: M=1.12, SD=0.47, n=95 vs CG [control group]: 1.19, SD=0.47, n=55; p=0.67) or student self-reports (M=1.20, SD=0.44, n=90 vs M=1.14, SD=0.36, n=42; p=0.84) at 1 week postintervention.[34] This finding was the same in the Bullying Literature Project—Moral Disengagement version (F(1, 80)=0.83, p=0.431), though only combined student-reported physical and emotional bullying estimates were available.[41]

Findings for Steps to Respect differed by type of rater. At the end of the first intervention year, the first evaluation of Steps to Respect[37] reported evidence of decreased bullying based on playground observation ($F(91.3)=5.02$, p<0.01) but not direct aggression based on student report ($F(68.7)=2.05$, p>0.05). The second evaluation of Steps to Respect[42] revealed a similar pattern. While teacher reports of physical bullying perpetration were less in intervention schools than in control schools at the end of the first intervention year (OR=0.61, $t(29)=-3.12$, p<0.01), student reports suggested no difference between schools on bullying perpetration ($t(29)=-1.06$). Moreover, in PATHS,[38] small positive effects of the intervention on student-reported aggression at the end of the first intervention year ($d=-0.048$, 95% CI −0.189 to 0.092) and at the start ($d=-0.064$, 95% CI −0.205 to 0.076) and end ($d=-0.048$, 95% CI −0.188 to 0.093) of the second intervention year gave way to a small deleterious intervention effect at the end of the third year ($d=0.082$, 95% CI −0.060 to 0.224). Opposite effects were found on teacher-reported aggression, with initially small, negative intervention effects at the end of the first ($d=0.036$, 95% CI −0.105 to 0.178) and start of the second intervention year ($d=0.035$, 95% CI −0.107 to 0.178) but progressively greater effects at the end of the second ($d=-0.005$, 95% CI −0.146 to 0.136) and the third ($d=-0.199$, 95% CI −0.338 to −0.060) intervention years.

In contrast, two evaluations showed consistently positive results across different measures. In Positive Action Chicago,[36] students reported lower counts of bullying behaviours (incidence rate ratio (IRR)=0.59, 95% CI

**Table 4** Measures used in included studies and effect estimates

| Evaluation | Measure | Notes | Effect estimate |
|---|---|---|---|
| Violence perpetration | | | |
| Reading, Writing, Respect and Reconciliation | Aggression | Frequency score on 13 aggressive behaviours assessed by teacher report in last month, including physical aggression and threatening of others | Key stage (KS)2<br>End of first year: regression-estimated $b$=0.02, SE=0.05, based on a 1–4 scale<br>End of second year: $d$=−0.21, p<0.05 |
| Bullying Literature Project | Physical bullying | Assessed by teacher and student report; mean of frequency scores relating to reports of violence | KS2<br>Teacher report: IG: M=1.12, SD=0.47, n=95 versus CG: 1.19, SD=0.47, n=55; p=0.67<br>Student report: 1.20, 0.44, n=90 versus 1.14, 0.36, n=42; p=0.84 |
| Bullying Literature Project—Moral Disengagement | Bullying | Assessed by student report; mean of frequency scores relating to physical and emotional bullying | KS2<br>No significant difference from time by treatment interaction: F(1, 80)=0.83, p=0.431 |
| Linking the Interests of Families and Teachers | Change in child physical playground aggression | Measured by observation; includes physical bullying by observed children | KS2<br>'Statistically significant' differences: $d$=−0.14 at mean, 1 SD and 2 SD above the preintervention mean |
| Promoting Alternative Thinking Strategies (PATHS) | Aggression | Assessed by teacher and student report; mean of frequency scores relating to verbal and physical aggression | KS2<br>Student report: decreased at the end of first year $d$=−0.048, 95% CI −0.189 to 0.092); start of second year (−0.064, 95% CI −0.205 to 0.076); end of second year (−0.048, 95% CI −0.188 to 0.093); but increased at the end of the third year (0.082, 95% CI− 0.060 to 0.224)<br>Teacher report: increased at the end of the first year (0.036, 95% CI −0.105 to 0.178), start of second year (0.035, 95% CI −0.107 to 0.178) but decreased at the end of the second year (−0.005, 95% CI − 0.146 to 0.136) and end of third year (−0.199, 95% CI −0.338 to −0.060) |
| Positive Action Chicago | Bullying | Student report: count of bullying behaviours relating to verbal or physical aggression behaviours in the past 2 weeks<br>Parent report: count of observed verbal or physical aggression behaviours in the past 30 days | KS2<br>Student report incidence rate ratio (IRR)=0.59, 95% CI 0.37 to 0.92<br>KS3<br>Student report: $d$=−0.39<br>Parent report: $d$=−0.31 |
| | Violence-related behaviours | Count of lifetime behaviours: carried a knife, threatened to cut or stab someone, cut or stabbed someone on purpose, been asked to join a gang, hung out with gang members, been a member of a gang | KS2<br>IRR=0.63, 95% CI 0.45 to 0.88<br>KS3<br>IRR=0.38, 95% CI 0.18 to 0.81, or $d$=−0.54 |
| Positive Action Hawaii | Count of violent behaviours | Teacher, student report | KS2<br>Teacher report: IRR=0.54, 90% CI 0.30 to 0.77<br>Student report: IRR=0.42, 90% CI 0.24 to 0.73 |
| | Cut or stabbed others | Student report, lifetime prevalence | KS2<br>OR=0.29, 90% CI 0.16 to 0.52 |
| | Shot another person | Student report, lifetime prevalence | KS2<br>OR=0.24, 90% CI 0.14 to 0.40 |
| | Physically hurts others | Teacher report | KS2<br>OR=0.61, 90% CI 0.38 to 0.97 |
| | Gets into a lot of fights | Teacher report | KS2<br>OR=0.63, 90% CI 0.47 to 0.84 |

Continued

**Table 4** Continued

| Evaluation | Measure | Notes | Effect estimate |
|---|---|---|---|
| Second Step | Physical aggression perpetration | Student report, endorse any fighting behaviours in the last 30 days | KS3<br>End of first year: OR=0.70, p<0.05<br>End of second year: OR=0.80, 95% CI 0.59 to 1.08<br>End of third year: β=0.005, SE=0.012 |
| | Sexual harassment and violence perpetration | Student report, endorse any verbal sexual violence or groping behaviours or forced sexual contact | KS3<br>End of first year: OR=1.04, p>0.05<br>End of second year: Illinois schools 0.72 (0.54, 0.95), Kansas schools 0.99 (0.71, 1.48) |
| Steps to Respect I | Bullying | Playground observation of students | KS2<br>Decrease in intervention group: *F*(91.3)=5.02, p<0.01 |
| | Direct aggression | Mean of student reported frequency scores of direct bullying | Decrease not significant in intervention group compared with control: *F*(68.7)=2.05, p>0.05 |
| Steps to Respect II | Bullying perpetration | Measured by student report; proportion of students with at least one bullying behaviour | KS2<br>Intervention group not significantly lower than control group: *t*(29)=−1.06 |
| | Physical bullying perpetration | Measured by teacher report; proportion of students with at least one physical bullying behaviour | KS2<br>Significantly less in intervention group: OR=0.61, *t*(29)=−3.12, p<0.01 |
| Youth Matters | Bullying | At least two or three times a month on at least one bullying behaviour | KS2<br>OR=0.85, 95% CI 0.29 to 1.47, p=0.585 |
| | Bully, victim or bully–victim | Classification of students based on questionnaire responses into one of three categories | Bully or bully–victim<br>KS2<br>End of first year IG: 21%, n=356 versus CG: 22%, n=392; end of second year 19%, n=244 versus 23%, n=293<br>KS3<br>Both groups 16%; IG n=283, CG n=289 |
| Violence victimisation | | | |
| Bullying Literature Project | Physical bullying | Assessed by teacher and student report; mean of frequency scores relating to reports of violence | KS2<br>Teacher report: IG: M=1.04, SD=0.23, n=95 versus CG: 1.04, SD=0.21, n=55; p=0.39<br>Student report: (1.35, 0.54, n=90 versus 1.43, 0.66, n=42; p=0.57 |
| Bullying Literature Project—Moral Disengagement | Bullying victimisation | Assessed by student report; mean of frequency scores relating to physical and emotional bullying | KS2<br>Student report: IG: M=1.76, SD=0.81 to M=1.60, SD=0.66, n=42 versus CG: M=1.23, SD=0.38 to M=1.38, SD=0.53, n=42; F(1, 80)=7.42, p=0.047 |
| Gatehouse | Bullying victimisation | Assessed by student report; any of being teased, having rumours spread about them, deliberate exclusion or experience of threats or violence | KS4<br>End of first year OR=1.03, 95% CI 0.86 to 1.26<br>End of second year OR=1.03, 95% CI 0.78 to 1.34<br>End of third year OR=0.88, 95% CI 0.68 to 1.13 |
| Learning to Read in a Healing Classroom | Victimisation | Assessed by student report; average of frequency scores of peer verbal and physical bullying | KS2<br>Weighted *d*=−0.01, SE=0.06 |
| PATHS | Victimisation | Assessed by student report; sum of frequency scores of victimisation in last 2 weeks | KS2<br>Increase at the end of the first intervention year (*d*=0.044, 95% CI −0.098 to 0.185); the start (0.074, 95% CI −0.067 to 0.216) and end (0.092, 95% CI −0.050 to 0.234) of the second year; and the end of the third year (0.089, 95% CI −0.053 to 0.231) |

Continued

**Table 4** Continued

| Evaluation | Measure | Notes | Effect estimate |
|---|---|---|---|
| Second Step | Peer victimisation | Student report, endorse any physical or verbal victimisation in last 30 days | KS3<br>End of first year OR=1.01, p>0.05<br>End of second year OR=0.94, 95% CI 0.75 to 1.18 |
| | Sexual harassment and violence victimisation | Student report, endorse any victimisation by verbal sexual violence or groping behaviours or forced sexual contact | KS3<br>End of first year OR=1.01, p>0.05<br>End of second year OR=0.91, 95% CI 0.72 to 1.15 |
| Steps to Respect I | Target of bullying | Playground observation of students | KS2<br>IG: M=0.9, SD=0.82 versus CG: M=1.01, SD=0.83; $F(72.4)=3.74$, p<0.10 |
| | Victimisation | Assessed by student report; mean of frequency scores for physical and verbal victimisation items | KS2<br>IG: M=0.80, SD=1.51 versus CG: M=0.86, SD=1.44; $F<1$ |
| Steps to Respect II | Victimisation | Assessed by student report; mean of frequency scores for physical and verbal victimisation items | KS2<br>IG: M=2.11, SD=1.03 versus CG: M=2.18, SD=1.06; $t(29)=-1.15$ |
| Youth Matters | Victimisation | Assessed by student report; mean of frequency scores for physical and verbal victimisation items, and also at least two or three times a month victimisation at least one bullying behaviour | KS2<br>difference=−0.171, SE=0.083, p=0.049; OR=0.61, p=0.098<br>KS3<br>Regression-estimated difference=−0.123, SE=0.068, p=0.08 |
| | Bully, victim or bully–victim | Classification of students based on questionnaire responses into one of three categories | Victim or bully–victim<br>KS2<br>No difference between groups<br>KS3<br>IG: 36%, n=283 versus CG: 45%, n=289 |

0.37 to 0.92) and of serious violence-related behaviours, including cutting or stabbing someone on purpose (IRR=0.63, 95% CI 0.45, 0.88). Findings from Positive Action Hawaii[43] were similar for student-reported violent behaviours (IRR=0.42, 90% CI 0.24 to 0.73) and teacher-reported violent behaviours (IRR=0.54, 90% CI 0.30, 0.77). For students in the fourth or fifth intervention year, intervention recipients were less likely to report cutting or stabbing someone (OR=0.29, 90% CI 0.16 to 0.52) or shooting someone (OR=0.24, 90% CI 0.14, 0.40). Teachers were less likely to report that students hurt others (OR=0.61, 90% CI 0.38, 0.97) or got into lots of fights (OR=0.63, 90% CI 0.47, 0.84).

However, in Youth Matters,[35] students in intervention schools were not less likely to report bullying perpetration (OR=0.85, 95% CI 0.29 to 1.47, p=0.585) after the second intervention year. Evaluators explored use of latent class analyses to classify intervention recipients as victims, bullies or bully–victims. Proportions of intervention and control recipients classified as bullies or bully–victims were not significantly different by study arm at the end of the first (IG: 21%, n=356 vs CG: 22%, n=392) or second (19%, n=244 vs 23%, n=293) intervention years.[44]

**Violence perpetration: KS3**

The three evaluations examining violence perpetration outcomes in KS3 had dissimilar results. At the end of

the sixth intervention year of Positive Action Chicago,[45] students receiving the intervention reported lower counts of violence-related behaviours than no treatment controls (IRR=0.38, 95% CI 0.18 to 0.81; equivalent to $d=-0.54$). Students also reported fewer bullying behaviours ($d=-0.39$), and parents reported that their children engaged in fewer bullying behaviours ($d=-0.31$). Significance values for these estimates were not presented, but both were supported by significant condition by time interactions in multilevel models, indicating that the intervention group showed an improved trajectory over time as compared with the control group. In contrast, after the third year from baseline in Youth Matters,[44] proportions of students were not different in the collective bully and bully–victim groups (both groups 16%; IG n=283, CG n=289). Findings for Second Step were reported at the end of the first, second and third years of intervention. At the end of the first school year, students in intervention schools had decreased odds of physical aggression (OR=0.70, p<0.05) but not sexual harassment and sexual violence perpetration (OR=1.04, p>0.05).[33] These findings did not hold to the end of the second school year for physical aggression (OR=0.80, 95% CI 0.59 to 1.08), but sexual harassment and sexual violence perpetration was significantly reduced in intervention schools in Illinois (OR=0.72, 95% CI 0.54, 0.95) but not Kansas (OR=0.99,

95% CI 0.71, 1.48).[32] At the end of the third school year, there were no direct effects of Second Step on sexual harassment perpetration (β=0.005, SE=0.012); findings for physical aggression were not available.[31]

## Violence victimisation: KS2

While the seven evaluations reporting outcomes in this KS were similar in follow-up period, they did not point to a clear effect. Students receiving the 'original' Bullying Literature Project were not different from their peers in physical victimisation by teacher report on individual students (IG: M=1.04, SD=0.23, n=95 vs CG: 1.04, SD=0.21, n=55; P=0.39) or student self-report (M=1.35, SD=0.54, n=90 vs M=1.43, SD=0.66, n=42; P=0.57) 1 week postintervention.[34] However, students receiving the Bullying Literature Project—Moral Disengagement version did report decrease in victimisation (both physical and emotional combined) after the intervention (IG: M=1.76, SD=0.81 to M=1.60, SD=0.66, n=42 vs CG: M=1.23, SD=0.38 to M=1.38, SD=0.53, n=42), with a significant time-by-treatment interaction in an analysis of variance (F(1, 80)=7.42, P=0.047).[41] PATHS measured student-reported victimisation using standardised mean differences, and found small, non-significant increases relative to the control arm at: the end of the first intervention year (d=0.044, 95% CI −0.098 to 0.185); the start (d=0.074, 95% CI −0.067, 0.216) and end (d=0.092, 95% CI −0.050, 0.234) of the second year; and the end of the third year (d=0.089, 95% CI −0.053, 0.231) of intervention implementation.[38] Steps to Respect, evaluated in two different trials, also found no differences in student-reported bullying victimisation at the end of the first intervention year in the first (IG: M=0.80, SD=1.51 vs CG: M=0.86, SD=1.44; F<1)[37] or second trial (M=2.11, SD=1.03 vs M=2.18, SD=1.06; t(29)=-1.15).[42] The first trial included playground observation at the end of the first intervention year, which was suggestive of lower levels in bullying victimisation, though these differences were marginally non-significant (M=0.9, SD=0.82 vs M=1.01, SD=0.83; F(72.4)=3.74, p<0.10).[37] Learning to Read in a Healing Classroom examined relational and physical victimisation after 1 year of intervention implementation and found no significant effect of the intervention (weighted d=−0.01, SE=0.06).[29 30] Finally, Youth Matters examined bullying victimisation through continuous and dichotomous measures. At the end of the second intervention year, the difference in log-transformed continuous scores suggested a decrease (difference=−0.171, SE=0.083, p=0.049), as did the difference in dichotomous scores (OR=0.61, p=0.098).[35] However, a latent class analysis that sought to describe transitions into, and out of, bullying victimisation did not suggest a difference between groups at this point.[44]

## Violence victimisation: KS3 and KS4

Intervention evaluations reporting violence victimisation outcomes in KS3 (Youth Matters,[44 46] Second Step[32 33] and Gatehouse[28]) and KS4 (Gatehouse[28]) suggested no evidence of effectiveness. In Youth Matters, differences in the log-transformed scores for bullying victimisation suggested a decrease in victimisation in intervention recipients as compared with controls, but this difference was not significant (regression-estimated difference=−0.123, SE=0.068, p=0.08).[46] However, at the end of the third intervention year, fewer students in the intervention than control group were members of the victim or bully–victim classes (36%, n=283 vs 45%, n=289).[44] Based on our own $\chi^2$ test, this difference was significant (p=0.029). In Second Step, neither peer victimisation (OR=1.01, p>0.5) nor sexual harassment and violence victimisation (OR=1.01, p>0.05) were different between students in intervention schools and control schools after the first intervention year.[33] This remained the case at the end of the second intervention year (peer victimisation: OR=0.94, 95% CI 0.75, 1.18); sexual victimisation: OR=0.91, 95% CI 0.72, 1.15).[32] Gatehouse,[28] which was implemented from year 9, found no evidence of a change in bullying victimisation at the end of the first (OR=1.03, 95% CI 0.86, 1.26)), second (OR=1.03, 95% CI 0.78, 1.34) or third (OR=0.88, 95% CI 0.68, 1.13) intervention years, which corresponded to the first 2 years of KS4.

## DISCUSSION

While the integration of academic and health education remains a promising model for the delivery of school-based health education, randomised evaluations were variable in quality and did not consistently report evidence of effectiveness in reducing violence victimisation or perpetration. Evidence was concentrated in KS2, with few evaluations in KS3 or KS4. Moreover, evidence was stronger in quantity and in quality for violence perpetration as compared with victimisation. Unfortunately, evaluations that measured perpetration did not always also measure victimisation, preventing a meaningful comparison of consistency of effects.

Few interventions showed consistent signals of effectiveness. Though a formal moderator analysis was not possible, certain intervention models appear more effective than others. Specifically, evaluations of Positive Action in both Chicago[45] and Hawaii[43] showed consistently positive results across diverse measures. This may reflect the involvement of the intervention developer, a factor often associated with improved intervention fidelity (although Positive Action was not unique in this respect among interventions included in our review). It may also reflect that Positive Action included classroom, whole-school and (in the Hawaii trial) external domain strategies delivered over multiple school years. Though Gatehouse[28] was similar to Positive Action in its focus on multiple systems, Gatehouse targeted adolescents, whereas Positive Action was delivered from KS2 and also included work with parents. Another possible explanation for our results is that effects for these interventions may take time to emerge. This is plausible given the developmental focus of many of these interventions, and evidence of links between early aggressive behaviour and later

violence.[4 5] For example, there was some evidence that effects on aggressive behaviour in 4Rs began to emerge after the second intervention year.[19] While findings were somewhat contradictory across different outcomes for PATHS, there was some evidence that teachers of intervention students reported less aggression in later years of the intervention.[38] Another key feature of Positive Action was the use of a model that linked academic and health education to developmental concerns. That is to say, this intervention focused on improvements in academic engagement and study skills both enhancing, and being enhanced by, student health and well-being; this was a feature of intervention activities and of the underlying theory of change. Moving forward, intervention strategies that combine multiple domains over several years and that use both subject-specific learning alongside linking to developmental concerns may be more effective than classroom-only interventions, single-year interventions or interventions that use literature alone; this should be a target for future research.

This systematic review has strengths and limitations. Identifying relevant studies was challenging often because of poor intervention description. We were unable to undertake meta-analysis or assessment of publication bias, though the preponderance of null results suggests that projects with non-significant findings are being published. Finally, the diversity of outcome measures and of raters precludes a complete and consistent picture of the effectiveness of these interventions via standardised measures. For example, measures that included physical violence and aggression were at times combined with verbal forms of interpersonal violence; while we preferred measures of physical violence and physical aggression, we included outcomes where these behaviours were included as part of a composite. Consistency and clarity in outcome reporting will be especially important as 'core outcome sets' become relevant in planning evaluations in public health and social science. Most studies focused on bullying, while evaluations of Positive Action[43 45] generally provided the most direct test of violent behaviours specifically.

Future research should seek to understand better the life course aspects of these interventions: that is, how does early school-based intervention impact later-life violent behaviours? From a policy perspective, it is clear that the integration of academic and health education, while possibly an effective intervention, will need to be considered alongside interventions involving other systems to prevent violence. Future evaluations will also contribute by considering the effects of integration in a diversity of ways and mechanisms of action for integration in different types of academic education. For example, contrasts between full and partial integration, which included evaluations did not address, could inform an understanding of how much integration is necessary to support health education messages.

**Author affiliations**
¹DECIPHer, School of Social Sciences, Cardiff University, Cardiff, UK
²Department of Social and Environmental Health Research, London School of Hygiene and Tropical Medicine, London, UK
³Cardiff University, Cardiff, UK
⁴DECIPHer, Bristol Medical School, University of Bristol, Bristol, UK
⁵EPPI-Centre, UCL Institute of Education, University College London, London, UK

**Acknowledgements** The authors acknowledge Ms Claire Stansfield for her assistance in designing and conducting the searches.

**Contributors** GJMT undertook study screening and selection, led the meta-analyses and drafted the initial manuscript. TT undertook study screening and selection, extracted data and contributed to drafting the initial manuscript. AF undertook study screening and selection. JT and RC provided methodological and substantive advice. CB undertook study screening and selection, extracted data and contributed to drafting the initial manuscript. All authors revised the manuscript and approved the final manuscript as submitted.

**Funding** This work was funded through National Institute for Health Research Public Health Research Programme grant 14/52/15. The work was undertaken with the support of The Centre for the Development and Evaluation of Complex Interventions for Public Health Improvement (DECIPHer), a UKCRC Public Health Research Centre of Excellence. Joint funding (MR/K0232331/1) from the British Heart Foundation, Cancer Research UK, Economic and Social Research Council, Medical Research Council, the Welsh Government and the Wellcome Trust, under the auspices of the UK Clinical Research Collaboration, is gratefully acknowledged.

**Disclaimer** The views expressed are those of the authors and not necessarily those of the NHS, the NIHR or the Department of Health and Social Care.

**Competing interests** None declared.

**Patient consent** Not required.

**Provenance and peer review** Not commissioned; externally peer reviewed.

**Data sharing statement** All data are publicly available.

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
