## [Reviewer comments · BMJ Open]

This paper was submitted to a another journal from BMJ but declined for publication following peer review. The authors addressed the reviewers' comments and submitted the revised paper to BMJ Open. The paper was subsequently accepted for publication at BMJ Open.

(This paper received three reviews from its previous journal but only two reviewers agreed to published their review.)

ARTICLE DETAILS

TITLE (PROVISIONAL)	Integration of academic and health education for the prevention of physical aggression and violence in young people: systematic review, narrative synthesis and intervention components analysis
AUTHORS	Melendez-Torres, G.J.; Tancred, Tara; Fletcher, Adam; Campbell, Rona; Thomas, James; Bonell, Christopher

VERSION 1 – REVIEW

REVIEWER	Simon Denny University of Auckland
REVIEW RETURNED	12-Dec-2017

GENERAL COMMENTS	Thank you for the opportunity to review this paper, which systematically reviews the evidence of the effectiveness on integrating academic and health education to reduce violence. Overall the paper is interesting and brings together 10 evaluations of eight interventions. The authors conclude that 'the evaluations were variable in quality and did not consistently report evidence of effectiveness in reducing violence' (pg 13). One of the strengths of this manuscript is the comprehensive search strategy which returned 76,979 references. (I did wonder how the authors managed to review all these abstracts!) Of these 702 were retained for full text screening leading to a final 8 interventions reported in 14 papers. One concern I have about this search strategy is in the definition of 'health education' and the aim of the paper to analyse 'interventions that integrate health and education' to prevent violence. There are a number of interventions that may not be captured in this review. For example, how do socio-emotional learning programs fit within this definition? And are they included in the search criteria? The authors also state that there have been no reviews in this area, but I wonder if this is due to how the authors have defined health interventions (for example Wilson SJ, Lipsey MW. School-based interventions for aggressive and disruptive behavior: update of a meta-analysis. Am J Prev Med 2007;33(2S):S130–S143.). So it would help if there was more clarity around how school based interventions were defined and exactly what was meant by 'health' related interventions. I would also like the authors to more clearly outline the theories of change and how these interventions may be effective. At present
---

	the introduction provides a list of possible areas but a more critical perspective and how the field has progressed would be welcome. For example, how socio-emotional interventions have moved beyond self-esteem building to a more nuanced view of how self-esteem operates. It would be fascinating but perhaps outside the scope of this review to see which areas of non-cognitive skill development that different programs focus on that has the best evidence of their effectiveness. A further strength of the analyses was the use of an 'intervention components analysis' but this does need further explanation. The results section was comprehensive and broken down by age group or school year groupings. But a number of the interventions were multi-year and cut across these groupings. I wondered if a more useful categorization would be by duration of program as there were large differences depending on the duration or multi-year requirement. The results section was difficult to follow at times due to number of alternative effect metrics. Is there any way this could be simplified or made clearer? Lastly, in reviewing the studies analyzed there were clear differences in the quality of studies that were not evident in the appraisal in Table 2. For example, the number of schools (or classrooms) randomised varied from 6 to 20. I did wonder if the smaller studies had sufficient power to answer their research questions. Furthermore some interventions stretched out over 7 years, while other appear to be one semester. While heterogeneous outcome measure precluded formal meta-analysis and I liked the narrative format, I did wonder how such big differences in quality of studies could allow them to be grouped together without further qualification.
--	---

REVIEWER	Daniel Romer Annenberg Public Policy Center University of Pennsylvania USA
REVIEW RETURNED	17-Jan-2018

GENERAL COMMENTS	This paper presents a review of eight school-based interventions that use various educational strategies to reduce interpersonal aggression and it attempts to determine their efficacy as reported in 14 studies. It is clearly helpful to provide a catalog of these programs and to assess the evidence regarding their effects. I have some suggestions for ways to make the review more helpful for practitioners and interventionists. First, titling the paper with evidence regarding violence prevention is a bit misleading. Research on aggression in youth has tended to conflate various forms of bullying and verbal hostility with violence, and this has served to muddy the interpretation of the results. It would be preferable to title this review as pertaining to the prevention of aggressive behavior in youth, some of which might extend to violence. A good deal of bullying will only involve verbal forms of aggression which are quite harmful and may lead to violence, but which on their own do not reach the level of actual violence. The authors have done a service in describing the various programs in Table 1. However, it would be helpful to try to classify the educational strategies used in the various programs with greater clarity. There are a few dimensions that could be highlighted, such as does the intervention involve the entire school or just the isolated class? Does it involve active engagement with
--

	learning materials, or just lecture and reading? Does it involve attempts to increase skills for dealing with interpersonal conflicts? Having more details such as these in Table 1 would make the review more useful. Some of these strategies are noted in Table 3, but many of them focus on details that may not matter much in regard to intervention efficacy; whereas the ones mentioned above do seem to be good candidates. This would also help to sharpen the Discussion of the findings which do not seem to provide much guidance as to why some programs worked better than others (e.g., page 13, lines 26-46). It would also help to indicate the age group targeted using the K system in the paper for each intervention and the age at which the program is initiated, the duration of the program, and the length of follow-up.
--	---

REVIEWER	Gina Martin University of St Andrews, United Kingdom
REVIEW RETURNED	06-Feb-2018

GENERAL COMMENTS	This research describes a systematic review of the integration of violence prevention in academic and health education. The systematic review protocol was previously registered on PROSPERO. The methods used are appropriate for the topic. The paper is well written, and the topic is of importance to public health. It is difficult to ascertain if the researchers answered their research questions as none were defined. Giving a clear set of research questions would benefit this manuscript; although they are implied throughout the paper. There are several points that need clarification or more detail in terms of the methods and results.  - Studies were included regardless of language. How was this conducted? For instance, were abstracts/papers translated or was a native speaker sought out? - On page 6, lines 18-25, the description of the study selection process would benefit from more detail. Were disagreements discussed before moving to the single reviewer screen? - Some discussion on the intervention components analysis (including a rationale for using this approach and details on how it was undertaken in this study) is needed in the methods section. - The results from studies use a variety of statistical methods. These should be better defined in the results section, for example in the text using statements such as 'the difference in means', etc. Also including each studies results in a table such as Table 1 would make it more straightforward to compare results. - The authour's state that poor intervention description was a challenge in identifying studies for inclusion. Were authours contacted to provide details if sufficient details were missing? - On the PRISMA diagram it is not stated why some reports were excluded from the 62 (the second box from the bottom).
---

REVIEWER	Erika E. Atienzo School of Health and Related Research, University of Sheffield, UK I declare the following competing interest: One of the co-authors of this manuscript (Rona Campbell) is listed as one of two line managers in a vacancy I might apply in the near future. The vacancy and therefore information about the managers, was
-----------------	---

	posted online after I had accepted to review this manuscript. I declare that I have no personal or institutional relationship with the mentioned researcher or the rest of the authors. I declare I have no other competing interest.
REVIEW RETURNED	11-Feb-2018

GENERAL COMMENTS	The aim of this study is to review evidence on the effectiveness of interventions including integration of academic and health education for reducing violence and describe the content of these interventions. The paper does not require a statistical review as a meta-analysis was not performed due to diversity in the measurement of the outcomes. The statistics used to present the results of the included papers are clear. I have some minor observations regarding the presentation of the methods and results sections, that if addressed by the authors the contribution of the manuscript could be improved. My main concerns relate to the interventions components analysis. Comments: Can the authors explain the rationale behind excluding interventions training teachers in classroom management? (page 6, first paragraph). It is not clear to me. On page 6, the authors present a 'study selection' section but also in this section they include the approach to analysis and synthesis. Therefore I suggest to include a separate sub-title indicating analysis and/or synthesis and to include the information presented in lines 33-55. On the results, there is no information describing how many (or which) interventions were classified in each of the key stages K1-K5. I am assuming no interventions for KS1 and KS5 were found? The authors could describe this result with more detail. I know that the full description of each intervention is provided in table 1 but it would be very helpful if each intervention is described in terms of the results of the components analysis, in a table or figure. Otherwise the relevance of this element of analysis gets lost and this is important considering that the aim of this paper is also to describe the content of the interventions. Providing a clearer description of the components of the interventions could help to think initially in some hypothetical links between intervention effects and intervention components. Also, this could provide insight to other researchers conducting research related to interventions. If slightly forced, each of the results of the components analysis could be included in table 1 next to the description of intervention, or in a separate figure. Maybe the description of each of the components that is presented in table 3 could be included as text, using table 3 instead to present the full results per intervention. Also related to the interventions components part, I found a bit difficult to understand the 'approach to integration' theme. Perhaps the authors could provide more explanation to understand what this theme is about. How would be classified an intervention that relied on other forms of delivery that are no books or written materials? For example, role play or role model or other visual materials not language-based (posters or pictures)? Why is the
---

	focus on literature or language-based more relevant than other forms of delivery? Was this the only form you found? Also related to the components analysis, the authors describe in the text that 'local development' consists in an approach to integration 'where teachers were encouraged to decide the most appropriate way to integrate activities into daily instruction' (page 8, lines 20-25). However, table 3 describes local development as: Did interventions support teachers to link health education across academic subjects in each school? I am not sure the above descriptions are measuring the same aspect. As explained before, presenting the results of the components analysis for each intervention could help in understanding what each of the components are measuring.
--	---

VERSION 1 – AUTHOR RESPONSE

Reviewer 1

Reviewer Name: Simon Denny

Institution and Country: University of Auckland

Thank you for the opportunity to review this paper, which systematically reviews the evidence of the effectiveness on integrating academic and health education to reduce violence.

Overall the paper is interesting and brings together 10 evaluations of eight interventions. The authors conclude that 'the evaluations were variable in quality and did not consistently report evidence of effectiveness in reducing violence' (pg 13).

One of the strengths of this manuscript is the comprehensive search strategy which returned 76,979 references. (I did wonder how the authors managed to review all these abstracts!) Of these 702 were retained for full text screening leading to a final 8 interventions reported in 14 papers. One concern I have about this search strategy is in the definition of 'health education' and the aim of the paper to analyse 'interventions that integrate health and education' to prevent violence. There are a number of interventions that may not be captured in this review. For example, how do socio-emotional learning programs fit within this definition? And are they included in the search criteria?

- **We included socio-emotional learning interventions where these integrated socio-emotional with academic learning (in terms of teaching which seamlessly integrates socio-emotional learning into academic lessons or teaching of discrete socio-emotional learning curricula which also provide academic learning elements). We included search terms, as described in Online File 1, that related to emotion.**

The authors also state that there have been no reviews in this area, but I wonder if this is due to how the authors have defined health interventions (for example Wilson SJ, Lipsey MW. School-based interventions for aggressive and disruptive behavior: update of a meta-analysis. *Am J Prev Med* 2007;33(2S):S130–S143.). So it would help if there was more clarity around how school based interventions were defined and exactly what was meant by 'health' related interventions.

- **We now clarify in the last paragraph of the introduction how our review is unique as compared to other reviews of school-based interventions.**
- **We also provide additional detail in the methods section regarding what was included and what was not included.**

I would also like the authors to more clearly outline the theories of change and how these interventions may be effective. At present the introduction provides a list of possible areas but a more critical perspective and how the field has progressed would be welcome. For example, how socio-emotional interventions have moved beyond self-esteem building to a more nuanced view of how self-esteem operates. It would be fascinating but perhaps outside the scope of this review to see which areas of non-cognitive skill development that different programs focus on that has the best evidence of their effectiveness.

- **We provide additional discussion in the introduction on our work summarising the findings of our in-press synthesis of reports of theories of change for the studies included in our review. Theories of change did not consistently explore which psychological constructs might mediate intervention effects on violence prevention.**
- **Because the evaluations were relatively few in number and heterogeneous with regard to other outcomes reported, we could not synthesise these other outcomes—but we agree that this would have been of substantial interest.**

A further strength of the analyses was the use of an ‘intervention components analysis’ but this does need further explanation. The results section was comprehensive and broken down by age group or school year groupings. But a number of the interventions were multi-year and cut across these groupings. I wondered if a more useful categorization would be by duration of program as there were large differences depending on the duration or multi-year requirement. The results section was difficult to follow at times due to number of alternative effect metrics. Is there any way this could be simplified or made clearer?

- **We appreciate that the diversity of alternative effect metrics is a complication of this review. In this particular case, this diversity is inextricably linked to our decision to undertake narrative synthesis rather than a meta-analysis of included effect sizes. We originally adopted the key stage-based categorisation of interventions in order to bring order to the narrative synthesis, and we have aimed to describe in what year post-baseline the relevant outcomes were measured. In particular, we believed that the use of a key stage framework would contribute to the policy relevance of our work, and is attuned to developmental particularities relevant at each stage of child and adolescent development.**

Lastly, in reviewing the studies analyzed there were clear differences in the quality of studies that were not evident in the appraisal in Table 2. For example, the number of schools (or classrooms) randomised varied from 6 to 20. I did wonder if the smaller studies had sufficient power to answer their research questions. Furthermore some interventions stretched out over 7 years, while other appear to be one semester. While heterogeneous outcome measure precluded formal meta-analysis and I liked the narrative format, I did wonder how such big differences in quality of studies could allow them to be grouped together without further qualification.

- **Evaluation size was not formally part of quality assessment, as it is not strictly a measure of quality. However, we agree that this range in size was worthy of mention and have now described it in the results section.**

Reviewer 2

Reviewer Name: Daniel Romer

Institution and Country: Annenberg Public Policy Center, University of Pennsylvania, USA

This paper presents a review of eight school-based interventions that use various educational strategies to reduce interpersonal aggression and it attempts to determine their efficacy as reported in

14 studies. It is clearly helpful to provide a catalog of these programs and to assess the evidence regarding their effects. I have some suggestions for ways to make the review more helpful for practitioners and interventionists.

First, titling the paper with evidence regarding violence prevention is a bit misleading. Research on aggression in youth has tended to conflate various forms of bullying and verbal hostility with violence, and this has served to muddy the interpretation of the results. It would be preferable to title this review as pertaining to the prevention of aggressive behavior in youth, some of which might extend to violence. A good deal of bullying will only involve verbal forms of aggression which are quite harmful and may lead to violence, but which on their own do not reach the level of actual violence.

- **As described in the methods section, we specifically included outcomes accounting for physical violence, preferring the ‘most direct’ estimates of physical violence. However, we appreciate that aggression is a relevant developmental construct. We have retitled the paper to describe ‘physical aggression and violence’.**

The authors have done a service in describing the various programs in Table 1. However, it would be helpful to try to classify the educational strategies used in the various programs with greater clarity. There are a few dimensions that could be highlighted, such as does the intervention involve the entire school or just the isolated class? Does it involve active engagement with learning materials, or just lecture and reading? Does it involve attempts to increase skills for dealing with interpersonal conflicts? Having more details such as these in Table 1 would make the review more useful. Some of these strategies are noted in Table 3, but many of them focus on details that may not matter much in regard to intervention efficacy; whereas the ones mentioned above do seem to be good candidates. This would also help to sharpen the Discussion of the findings which do not seem to provide much guidance as to why some programs worked better than others (e.g., page 13, lines 26-46).

- **We have endeavoured to present more detail of interventions using the component scheme described in Table 3.**
- **Because the focus of our review was on integration of academic and health education, we did not seek specifically to analyse in detail components of interventions, such as specific types of interpersonal skills. We have, however, presented additional information regarding e.g. how literature lessons were delivered, or how education was linked to developmental concerns.**
- **We have included additional information in the discussion on why some interventions were better than others. Because so few interventions provided consistent signals of effectiveness, it is difficult to consider why some worked better than others.**

It would also help to indicate the age group targeted using the K system in the paper for each intervention and the age at which the program is initiated, the duration of the program, and the length of follow-up.

- **Where this information was not already provided in Table 1, we have described it.**

Reviewer 3

Reviewer Name: Gina Martin

Institution and Country: University of St Andrews, United Kingdom

This research describes a systematic review of the integration of violence prevention in academic and health education. The systematic review protocol was previously registered on PROSPERO. The

methods used are appropriate for the topic. The paper is well written, and the topic is of importance to public health.

It is difficult to ascertain if the researchers answered their research questions as none were defined. Giving a clear set of research questions would benefit this manuscript; although they are implied throughout the paper. There are several points that need clarification or more detail in terms of the methods and results.

- **We have presented research questions at the end of the introduction section.**

-Studies were included regardless of language. How was this conducted? For instance, were abstracts/papers translated or was a native speaker sought out?

- **Where relevant we organised translation of studies in order to screen them but none of our included studies was written in a language other than English.**

-On page 6, lines 18-25, the description of the study selection process would benefit from more detail. Were disagreements discussed before moving to the single reviewer screen?

- **We discussed disagreements and this has now been clarified.**

-Some discussion on the intervention components analysis (including a rationale for using this approach and details on how it was undertaken in this study) is needed in the methods section.

- **We have now provided additional information in the methods section.**

-The results from studies use a variety of statistical methods. These should be better defined in the results section, for example in the text using statements such as 'the difference in means', etc. Also including each studies results in a table such as Table 1 would make it more straightforward to compare results.

- **We have now summarised results in Table 4. Where effect size metrics were not clear in the text of the Results section, we have ensured that this is now the case.**

- The author's state that poor intervention description was a challenge in identifying studies for inclusion. Were authors contacted to provide details if sufficient details were missing?

- **Yes, we contacted authors for details where necessary and this is now reported in the Methods section.**

- On the PRISMA diagram it is not stated why some reports were excluded from the 62 (the second box from the bottom).

- **We discuss in the first paragraph of the results section that the 62 reports contributed to the broader review. We have now edited this in the PRISMA diagram.**

Reviewer 4

Reviewer Name: Erika E. Atienzo

Institution and Country: School of Health and Related Research, University of Sheffield, UK

The aim of this study is to review evidence on the effectiveness of interventions including integration of academic and health education for reducing violence and describe the content of these interventions. The paper does not require a statistical review as a meta-analysis was not performed due to diversity in the measurement of the outcomes. The statistics used to present the results of the included papers are clear.

I have some minor observations regarding the presentation of the methods and results sections, that if addressed by the authors the contribution of the manuscript could be improved. My main concerns relate to the interventions components analysis.

Comments:

Can the authors explain the rationale behind excluding interventions training teachers in classroom management? (page 6, first paragraph). It is not clear to me.

- **We excluded these interventions when they did not involve the teaching of a defined curriculum that aimed to integrate academic and health education. Classroom management training interventions therefore were excluded since these do not teach a specific curriculum.**

On page 6, the authors present a 'study selection' section but also in this section they include the approach to analysis and synthesis. Therefore I suggest to include a separate sub-title indicating analysis and/or synthesis and to include the information presented in lines 33-55.

- **We have now done this.**

On the results, there is no information describing how many (or which) interventions were classified in each of the key stages K1-K5. I am assuming no interventions for KS1 and KS5 were found? The authors could describe this result with more detail.

- **We have now described this specifically in the first paragraph headlining section 'Intervention effects'.**

I know that the full description of each intervention is provided in table 1 but it would be very helpful if each intervention is described in terms of the results of the components analysis, in a table or figure. Otherwise the relevance of this element of analysis gets lost and this is important considering that the aim of this paper is also to describe the content of the interventions. Providing a clearer description of the components of the interventions could help to think initially in some hypothetical links between intervention effects and intervention components. Also, this could provide insight to other researchers conducting research related to interventions. If slightly forced, each of the results of the components analysis could be included in table 1 next to the description of intervention, or in a separate figure. Maybe the description of each of the components that is presented in table 3 could be included as text, using table 3 instead to present the full results per intervention.

- **Our reworked Table 3 now presents the information as requested, including how each intervention maps onto the different aspects of the components analysis.**

Also related to the interventions components part, I found a bit difficult to understand the 'approach to integration' theme. Perhaps the authors could provide more explanation to understand what this theme is about. How would be classified an intervention that relied on other forms of delivery that are no books or written materials? For example, role play or role model or other visual materials not language-based (posters or pictures)? Why is the focus on literature or language-based more relevant than other forms of delivery? Was this the only form you found?

- The reviewer is correct to note that a subgroup of interventions described targeting language-based lessons specifically. We did not find other approaches using, for example, maths or science lessons explicitly to integrate academic learning.
- To clarify this, we have provided additional explanation in the text, including discussion of how these components were manifested in the included interventions

Also related to the components analysis, the authors describe in the text that ‘local development’ consists in an approach to integration ‘where teachers were encouraged to decide the most appropriate way to integrate activities into daily instruction’ (page 8, lines 20-25). However, table 3 describes local development as: Did interventions support teachers to link health education across academic subjects in each school? I am not sure the above descriptions are measuring the same aspect. As explained before, presenting the results of the components analysis for each intervention could help in understanding what each of the components are measuring.

- We have clarified this wording by bringing the text in line with the table.

VERSION 2 – REVIEW

REVIEWER	Daniel Romer University of Pennsylvania Philadelphia, PA USA
REVIEW RETURNED	08-May-2018

GENERAL COMMENTS	This revision responds to many of the concerns raised in the first submission. It now provides a summary of the intervention components that were featured in each program (Table 3) and a table (4) that summarizes the outcomes. As a reader, the sense I get is that Positive Action is the most successful overall; however, several others were successful in reducing aggressive behavior as measured in perpetration. There is a noticeable lack of evidence for any of the interventions when assessed in terms of victimization. It would help readers to comment on this as the discrepancy is striking. One would expect parallel patterns of outcomes. Apparently Positive Action did not assess victimization, so we can't assess the consistency of measurement for this program. There are still some places in the paper where things are confusing. For example, the intervention component labelled “linking to developmental concerns” (page 10, lines 31-51) is not very clearly described. How do the programs with this feature enhance the “interrelationships between academic success and broader development, health and wellbeing.”? Similar questions arise with regard to the meaning of “local development.” (lines 22-26). Table 3 does not help very much in explaining these terms. Page 6, lines 46-51: how can one include outcomes that are both physical and non-physical violence? This raises a concern mentioned in the first review that aggression and violence are not necessarily the same thing. Page 6, line 33: Our also definition? Page 7, lines 5-7: this was said earlier on page 6.
--

	Page 8, lines 2-4: interventions provided in included interventions? Also, what are “key domains of relevance in understanding the integration of academic and health education?” Page 14, lines 29-31: how can one have decreased odds of physical aggression but not sexual harassment and violence perpetration? Page 15, lines 51-53: what is the latent class analysis” Page 17, lines 38-47: here again the discussion of developmental concerns is not very clearly articulated. This seems important, but what makes a program sensitive to developmental concerns? Isn't education in general sensitive to developmental concerns?
--	---

REVIEWER	Gina Martin University of St Andrews UK
REVIEW RETURNED	04-May-2018

GENERAL COMMENTS	The authors have done an excellent job addressing the reviewers comments. The manuscript is improved following this revision.
---

REVIEWER	Erika E. Atienzo The University of Sheffield, UK
REVIEW RETURNED	16-May-2018

GENERAL COMMENTS	I was asked to review the contents of this paper specifically in relation to the analysis. Therefore, I raised a few observations regarding this in the first version of the manuscript. The authors have addressed all the concerns I pointed out during the first review, including adding a new table, and I believe that the paper has been improved. I have no further observations.
---

VERSION 2 – AUTHOR RESPONSE

Reviewer 2

Reviewer Name: Daniel Romer

Institution and Country: University of Pennsylvania

Philadelphia, PA USA

This revision responds to many of the concerns raised in the first submission. It now provides a summary of the intervention components that were featured in each program (Table 3) and a table (4) that summarizes the outcomes.

As a reader, the sense I get is that Positive Action is the most successful overall; however, several others were successful in reducing aggressive behavior as measured in perpetration. There is a noticeable lack of evidence for any of the interventions when assessed in terms of victimization. It would help readers to comment on this as the discrepancy is striking. One would expect parallel patterns of outcomes. Apparently Positive Action did not assess victimization, so we can't assess the consistency of measurement for this program.

- **We agree that this is an important point. We now describe this in the first paragraph of the discussion.**

- **‘Moreover, evidence was stronger in quantity and in quality for violence perpetration as compared to victimisation. Unfortunately, evaluations that measured perpetration did not always also measure victimisation, preventing a meaningful comparison of consistency of effects.’**

There are still some places in the paper where things are confusing. For example, the intervention component labelled “linking to developmental concerns” (page 10, lines 31-51) is not very clearly described. How do the programs with this feature enhance the “interrelationships between academic success and broader development, health and wellbeing.”? Similar questions arise with regard to the meaning of “local development.” (lines 22-26). Table 3 does not help very much in explaining these terms.

- **We have substantially reworked this paragraph to provide additional information, and to draw links between programmes.**
- **For example, to better describe interventions that link to developmental concerns, we have noted that ‘A third approach was *linking to developmental concerns*, emphasising not so much the comprehensive integration of academic and health education but rather the interrelationships between academic success and broader development, health and wellbeing. These interventions viewed academic education through a ‘health’ lens, in addition to viewing health education through an ‘academic’ lens. From a conceptual perspective, this meant that the interrelationships between academic achievement and student health and wellbeing were emphasised in theories of change. From a practical perspective, this meant that interventions paired activities such as study skills lessons with social-emotional learning (e.g., in PATHS). For example, the theory of change underlying Gatehouse related to the creation of healthy social milieus in schools that would also support academic attainment; practically, this manifested as enhancement of academic lessons to improve interpersonal skills and emotional regulation. Similarly, Positive Action tied together individual student attainment with student health and wellbeing in their theory of change, with lessons focused on problem-solving and goal-setting, among other topics.’**

Page 6, lines 46-51: how can one include outcomes that are both physical and non-physical violence? This raises a concern mentioned in the first review that aggression and violence are not necessarily the same thing.

- **This is a problem commonly encountered in systematic reviews of complex interventions; namely, that outcome measures may address as part of a composite the behaviours of proximal interest for the review. To address this, we now clarify that we preferred direct measures of physically violent and aggressive behaviours, but included outcomes where these were composites as part of measures of interpersonal violence which might include items on verbal forms of interpersonal violence, but not items such as damage to property.**
- **To address this concern as part of the first revision, we altered the title to include physical violence and physical aggression.**
- **In this revision, we additionally discuss outcome measurement as a limitation, including in this specific respect: ‘For example, measures that included physical violence and aggression were at times combined with verbal forms of interpersonal violence; while we preferred measures of physical violence and physical aggression, we included outcomes where these behaviours were included as part of a composite.’**

Page 6, line 33: Our also definition?

- **We have now edited this.**

Page 7, lines 5-7: this was said earlier on page 6.

- **We have removed the repetition and moved the remaining sentence to the first paragraph under 'Inclusion and exclusion'.**

Page 8, lines 2-4: interventions provided in included interventions? Also, what are "key domains of relevance in understanding the integration of academic and health education?"

- **We appreciate that the original text could have been clearer. This has now been edited to read:**
- **'Intervention descriptions were rarely detailed enough to permit 'deep' engagement with the specific content of the interventions provided in included evaluations. The intervention components analysis identified overarching domains that accounted for similarities and differences between interventions in their integration of academic and health education, and developed within each domain a set of overlapping categories that described these similarities and differences between interventions within each domain.'**

Page 14, lines 29-31: how can one have decreased odds of physical aggression but not sexual harassment and violence perpetration?

- **We now have edited to note that this outcome refers to 'sexual harassment and sexual violence perpetration', which resolves this apparent contradiction. We regret the lack of clarity in the original text.**

Page 15, lines 51-53: what is the latent class analysis?"

- **We have now clarified this reference to an exploratory analysis undertaken on the experimental data on Youth Matters.**
- **'However, a latent class analysis that sought to describe transitions into, and out of, bullying victimisation did not suggest a difference between groups at this point.'**

Page 17, lines 38-47: here again the discussion of developmental concerns is not very clearly articulated. This seems important, but what makes a program sensitive to developmental concerns? Isn't education in general sensitive to developmental concerns?

- **We have now added text to clarify this link.**
- **'That is to say, this intervention focused on improvements in academic engagement and study skills both enhancing, and being enhanced by, student health and wellbeing; this was a feature of intervention activities and of the underlying theory of change.'**

Reviewer 4

Reviewer Name: Erika E. Atienzo

Institution and Country: The University of Sheffield, UK

I was asked to review the contents of this paper specifically in relation to the analysis. Therefore, I raised a few observations regarding this in the first version of the manuscript. The authors have addressed all the concerns I pointed out during the first review, including adding a new table, and I believe that the paper has been improved. I have no further observations.

- **Thank you for your comments.**